# Retrospective Study of the Prevalence and Associated Factors of Gallbladder Polyps among Residents of Two Korean Cities

**DOI:** 10.3390/jcm13082290

**Published:** 2024-04-15

**Authors:** Oh-Sung Kwon, Young-Kyu Kim, Hyeon Ju Kim

**Affiliations:** 1Department of Medical Information, Jeju National University Hospital, Jeju-si 63241, Republic of Korea; dhtjd@naver.com; 2Department of Surgery, Jeju National University Hospital, Aran 13gil 15, Jeju-si 63241, Republic of Korea; 3Department of Family Medicine, Jeju National University Hospital, Aran 13gil 15, Jeju-si 63241, Republic of Korea; fmhjukim@hanmail.net

**Keywords:** gallbladder polyps, risk factor, alcohol consumption, age, Jeju Island

## Abstract

**Background/Aims:** Well-known risk factors for gallbladder polyps include metabolic syndrome, age, and dyslipidemia. Jeju Island is approximately 80 km from the Korean peninsula and is divided into two administrative regions (Jeju City and Seogwipo City), with Mount Halla intervening in the center. Jeju City has higher employment and birth rates than Seogwipo City. Age and alcohol consumption differ between the two regions, and these factors may affect the prevalence of gallbladder polyps (GBPs). Therefore, we investigated the prevalence of GBPs and compared various factors, including alcohol consumption habits and age, associated with GBPs among residents in the two regions. **Methods:** This study included 21,734 residents who visited the Health Screening and Promotion Center of Jeju National University Hospital between January 2009 and December 2019. We investigated the prevalence and associated factors of GBPs among residents of Jeju City and Seogwipo City. **Results:** The prevalence of GBPs in Jeju City and Seogwipo City was 9.8% and 8.9% (*p* = 0.043), respectively. The mean age and rate of high-risk alcohol intake were higher in Seogwipo City. The mean body mass index and levels of fasting blood glucose, total cholesterol, low-density lipoprotein cholesterol, aspartate aminotransferase, gamma-glutamyl transferase, and alkaline phosphatase were lower in Jeju City. **Conclusions:** This study demonstrated a significant difference in GBP prevalence between the two regions of Jeju Island. Age and alcohol consumption might contribute to this difference; however, further prospective cohort studies are warranted to confirm our findings.

## 1. Introduction

Gallbladder polyps (GBPs) are intraluminal lesions of the gallbladder that are typically asymptomatic. The frequency of GBP detection, and, therefore, the apparent incidence of GBPs, has been on the rise, owing to the increased frequency of abdominal ultrasound examinations being performed as part of general medical checkups [1]. GBP prevalence estimates range between 2.2% and 9.5%, varying with race and geographic location [2,3]. Previous research in Republic of Korea determined that the prevalence of GBP in 2010–2012 was higher than that in 2002–2004 [1].

Gallstones identified using abdominal ultrasonography typically do not require treatment unless there are accompanying symptoms. However, GBPs, which appear as small nodules within the gallbladder, require close monitoring because the possibility of malignancy cannot be excluded [4]. Regular follow-up checks or surgical treatments, such as cholecystectomy, are necessary for GBPs that exceed a specific size threshold, incurring continuous medical expenses. Additionally, medical concerns persist when a patient with a GBP >1 cm in diameter refuses cholecystectomy, given the risk of cancer development before the next ultrasonographic examination.

Jeju Island, located approximately 80 km south of the Korean Peninsula, was formed by volcanic eruptions. Its terrain is not suitable for paddy farming because of excessively rapid water drainage. Thus, miscellaneous grains such as buckwheat, millet, sorghum, and barley have been traditionally cultivated and consumed as the main staples on the island. Additionally, seafood is a major component of the island’s diet [5]. Jeju Island is divided into two administrative regions including Jeju City and Seogwipo City, with Mount Halla intervening at the center. Until the 1990s, road and transportation conditions were not favorable, making the movement of people and goods a challenge. Jeju City has an international airport and an international passenger ship terminal, which facilitate the movement of people and goods with greater ease than that which is possible in Seogwipo City (Figure 1). Thus, relative to Seogwipo City, Jeju City has become a more attractive destination for migrants and younger individuals because of its more vibrant commercial activities [6]. In 2020, >70% of the island’s inhabitants resided in Jeju City [7]. In contrast, Seogwipo City had a higher percentage of individuals aged ≥65 years compared with Jeju City in the same year [7], along with a greater proportion of people engaged in agriculture or fisheries [6].

Well-known risk factors for GBP development and progression include the male sex, age, dyslipidemia, and non-alcoholic fatty liver disease [1,8,9]. Numerous cohort studies suggest that the prevalence of GBP increases with age [10,11]. However, this finding is not universal, as some studies report conflicting results, leaving the association between advancing age and GBP prevalence a subject of ongoing debate [12,13]. Dietary factors can lead to variations in blood cholesterol levels, which in turn affect cholesterol concentrations in the gallbladder and the development of cholesterol polyps. Studies have shown that alcohol consumption may exert a protective effect against GBP formation by decreasing biliary cholesterol saturation [14,15]. Blood alcohol stimulates cholecystokinin secretion in the pancreas, counteracting cholesterol GBP formation by increasing gallbladder motility [16,17]. However, a Taiwanese group demonstrated alcohol consumption to be an independent risk factor for GBP [18].

We previously reported that the prevalence of gallstone disease was significantly lower among natives than among migrants on Jeju Island [5]. We suggested that the difference might be attributed to the distinct eating and drinking habits of these groups. A recent local survey, reported in newspapers, indicated that approximately 25% of Jeju residents are migrants from mainland Korea, with a higher proportion of migrants and natives residing in Jeju City than in Seogwipo City [19]. We hypothesized that Seogwipo City residents (SRs) are more likely to adhere to a traditional diet and that a considerable number are high-risk alcohol consumers. These dietary and drinking habits, particularly the consumption of cereals and seafood, may influence the development of GBPs, leading to a disparity in GBP prevalence between Jeju City and Seogwipo City.

Therefore, the primary objective of this study was to examine the variance in GBP prevalence between Jeju City residents (JRs) and SRs. The secondary objective was to analyze the factors associated with GBP prevalence in both cities.

## 2. Methods

### 2.1. Residents

A total of 23,468 participants who visited the Health Screening and Promotion Center of Jeju National University Hospital between January 2009 and December 2019 were screened. Of these, 1734 participants were excluded because they were not domiciled on Jeju Island (*n* = 235), were younger than 20 years old (*n* = 25), refused consent, returned incomplete questionnaires (*n* = 943), or underwent operations that could affect GBP prevalence or the presence of the gallbladder (*n* = 531). The operations were as follows: gastrectomy (*n* = 23); hepatectomy with cholecystectomy for liver cancer (*n* = 31), GB cancer (*n* = 8), or bile duct cancer (*n* = 6); and cholecystectomy for GBP (*n* = 87) or cholecystitis (*n* = 376). Therefore, 21,734 residents of Jeju Island participated in this study (Figure 2). This study categorized the residents of Jeju Island who visited the Health Screening and Promotion Center at the Jeju National University Hospital for medical checkups according to their places of residence into JRs and SRs. JRs and SRs were defined as residents domiciled in Jeju City and Seogwipo City at the respective times of their medical checkups. The hospital’s institutional review board reviewed and approved this study (IRB number 2021-11-013).

### 2.2. Questionnaire

The residents were requested to fill out a questionnaire, which included the following items: home address, mobile or telephone number, medical history (specifically, history of hypertension, diabetes mellitus, hyperlipidemia, and related medication history), and alcohol consumption (type of alcohol, number of glasses per occasion, and frequency per week). Objective and graded questions and questionnaire items were used to minimize self-reporting bias. No identity-revealing information about the questionnaire respondents or distributors was collected. To minimize recall bias, information about participants was not collected; this information was provided only when participants could not understand the questionnaire items.

### 2.3. Diagnosis of GBPs

Ultrasound examinations were conducted by radiologists using an IU22 high-resolution ultrasonography system (Koninklijke Philips Electronics N.V., Amsterdam, The Netherlands). Ultrasound scans were conducted after the participants fasted for over 8 h. GBPs were detected and diagnosed by ultrasonography when they were fixed hyperechoic masses protruding from the GB wall into the lumen. They were characterized by a lack of shift with positional changes. The number and maximal diameters of polypoid lesions were documented.

### 2.4. Definition of Physical Activity

Physical activity levels were evaluated using the participants’ questionnaires according to the World Health Organization (WHO)’s Global Recommendations on Physical Activity for Health 2010 [20]. When participants conducted vigorous-intensity activity for >75 min or moderate-intensity aerobic physical activity for >150 min per week, with aerobic activity lasting >10 min per session, they were defined as physically active.

### 2.5. Definitions of High-Risk Alcohol Drinkers and Metabolic Syndrome

Data regarding the participants’ alcohol consumption were collected using the questionnaire. We calculated pure alcohol consumption (type of alcohol × glasses × specific gravity of alcohol) and converted it to a Korean standard drink (7 g of pure alcohol/glass). High-risk alcohol drinkers were men who consumed ≥7 drinks of alcohol and women who consumed ≥5 drinks, with at least two alcohol-drinking occasions per week [21].

This study followed the definition of metabolic syndrome in the revised National Cholesterol Education Program criteria [22]. We diagnosed participants with metabolic syndrome when they met ≥3 of the following criteria: (1) waist circumference ≥90 cm for men or ≥80 cm for women using the Asian-Pacific criteria outlined by the International Association for the Study of Obesity and the International Obesity Task Force of World Health Organization [23]; (2) triglycerides ≥150 mg/dL or antidyslipidemic medication use; (3) high-density lipoprotein cholesterol <40 mg/dL in men or <50 mg/dL in women or antihypertensive medication use; (4) high blood pressure ≥130/85 mmHg or antihypertensive medication use; and (5) high fasting blood glucose ≥100 mg/dL or diabetes medication use (oral hypoglycemic agents or insulin).

### 2.6. Physical Examination

The participants’ heights and weights were measured (GL-150R, G-Tech International Co., Uijeongbu-si, Gyeonggi-do, Republic of Korea) without shoes and with a light gown. The body mass index was calculated using a participant’s height and weight (kg/m^2^). Sex and age were obtained from medical records. Venous blood samples were collected after 8 h of fasting. Aspartate aminotransferase, alanine aminotransferase, gamma-glutamyltransferase, alkaline phosphatase, total cholesterol, high-density lipoprotein cholesterol, low-density lipoprotein, triglycerides, and fasting blood glucose levels were measured using venous blood samples.

### 2.7. Statistical Analysis

We compared the prevalence and associated factors of GBPs using Student’s *t*-test for continuous variables and the chi-square test for categorical variables between the two groups (JR group vs. SR group). Statistical significance was set at *p*-value < 0.05. All statistical analyses were performed using PASW Statistics for Windows, version 18.0 (SPSS Inc., Chicago, IL, USA).

## 3. Results

### 3.1. Overall and Annual Prevalence of GBPs

Among the 21,734 participants, 11,872 (54.6%) were men and 9862 (45.4%) were women. There were 16,090 JRs and 5644 SRs. GBPs were diagnosed in 2079 participants, with an overall prevalence of 9.6%. The GBP prevalence in the JR and SR groups was 9.8% (*n* = 1577) and 8.9% (*n* = 502), respectively, with a statistically significant intergroup difference. The annual prevalence was lowest in 2018 (6.8%) and highest in 2014 (14.1%) during the study period (Figure 3). The annual prevalence tendencies of GBP in the JR and SR groups were similar. There was no correlation between the annual prevalence among residents of Jeju Island (*r* = 0.004, *p* = 0.514), JRs (*r* = 0.002, *p* = 0.804), SRs (*r* = 0.013, *p* = 0.317), or the study period.

### 3.2. Comparisons of Factors Affecting GBPs

The central obesity prevalence; high blood pressure prevalence; mean body mass index; and mean levels of fasting blood glucose, total cholesterol, low-density lipoprotein cholesterol, aspartate aminotransferase, gamma-glutamyltransferase, and alkaline phosphatase were significantly higher in the SR group than in the JR group. The prevalence of GBPs (9.8%) was significantly higher in the JR group than that in the SR group (8.9%). The proportion of high-risk alcohol drinkers and the mean age were significantly higher in the SR group than in the JR group (Table 1).

### 3.3. GBP Prevalence According to Age

To investigate whether age could affect the prevalence of GBPs, we analyzed the prevalence of GBPs according to age groups (≥60 vs. <60 years). Among participants aged ≥60 years, the GBP prevalence in the JR and SR groups was 9.1% and 8.1%, respectively, compared with 10.7% and 10.1%, respectively, among participants aged <60 years (*p* < 0.001). Among participants aged ≥60 years, the rates in the JR and SR groups were 56.8% and 52.3%, respectively, compared with 43.2% and 47.7% (*p* < 0.001), respectively.

### 3.4. Prevalence of GBPs and Proportions of High-Risk Alcohol Drinkers According to Age Group by Decade

To investigate whether high-risk alcohol drinking could affect GBP prevalence, we analyzed GBP prevalence and the proportions of high-risk alcohol drinkers according to age (by decade). The GBP prevalence in the JR and SR groups was, respectively, 4.9% and 2.3% among participants in their 20s, 9.8% and 7.9% among participants in their 30s, 10.8% and 11.8% among participants in their 40s, 11.6% and 10.1% among participants in their 50s, 10.2% and 9.2% among participants in their 60s, and 7.6% and 6.9% among participants aged ≥ 70 years. A significant difference in GBP prevalence was noted between participants in their 20s and 50s (Figure 4). The proportions of high-risk alcohol drinkers in the JR and SR groups were 35.9% and 47.9% in the 30s age group (*p* = 0.009), 36.3% and 41.5% in the 40s age group (*p* = 0.070), 33.4% and 40.3% in the 50s age group (*p* < 0.001), 29.4% and 29.5% in the 60s age group (*p* = 0.986), and 17.6% and 16.3% in the ≥ 70-year age group (*p* = 0.662), respectively (Figure 5).

## 4. Discussion

GBP prevalence has been reported to differ among regions. Previous studies conducted in the West have reported GBP prevalence between 1.0% and 6.9% [24,25], compared with 2.2% to 9.5% in Asia [1,2,26,27], and 2.9% to 9.9% in Korea specifically, depending on the region and target group [2,3]. In this study, the total prevalence was 9.6%, which was similar to the domestic GBP rate that was reported in a previous study [2,25].

Lee et al. [1] reported that the prevalence of GBPs in Korea continuously increased from 2002 to 2012. The authors stated that this increase was not associated with an increase in the prevalence of GBPs due to metabolic disorders, such as obesity and dyslipidemia; rather, it was associated with increased detection by ultrasonographic examination. We also investigated the correlation between GBP prevalence and the study period in the JR and SR groups for 11 years from the beginning of 2009 through the end of 2019 and found no statistically significant correlation (Figure 3). Despite employing ultrasonography equipment with the latest technological advancements throughout the study period, we did not observe a significant increase in GBP prevalence. Therefore, the prevalence of GBPs did not increase with improvements in ultrasonographic technology and resolution during the study period.

The highest and lowest prevalence of GBPs were observed in 2014 and 2018, respectively. We inferred that the variation was not attributable to changes in the actual prevalence of GBPs among residents or to the resolution of the ultrasound equipment but rather to the radiologists conducting the examinations. Ultrasound examinations were carried out by radiologists on an annual contract with the Health Screening and Promotion Center. The detection rate of GBPs could vary depending on the extent to which radiologists are encouraged to identify polypoid lesions or to distinguish GBPs from gallbladder stones, regardless of the ultrasound equipment’s resolution capabilities. Cross-validation is essential to minimize errors associated with interobserver variability.

The GBP prevalence in the JR and SR groups was 9.8% and 8.9%, respectively, representing a statistically significant difference. The known risk factors for GBP development, such as metabolic syndrome, mean high-density lipoprotein cholesterol levels, and mean triglyceride levels [28], were not significantly different between the two groups. The mean body mass index and fasting blood sugar, total cholesterol, and low-density lipoprotein cholesterol levels were lower in the JR group. Interestingly, the mean age and rate of high-risk alcohol consumption were higher in the SR group. These two factors may have exerted a decisive influence sufficient to offset the impacts of other known risk factors for GBP development.

The pathogenetic mechanisms underlying the development of GBPs are not well understood. However, the general hypothesis is that cholesterol levels in the blood or bile increase, causing concentrated cholesterol to create deposits in the bile. This, combined with gallbladder dysmotility, directly influences the formation of cholesterol polyps. In previous studies investigating the correlation between GBP prevalence and age, the prevalence of GBPs increased until the 50s but decreased after the 60s [2,29,30,31]. This phenomenon can be explained by decreased digestive ability and poor dental conditions associated with advanced age. High-fat dietary consumption tends to decrease among older adults, owing to problems with digestion and dentition [32]. Consequently, blood lipid metabolism improves, decreasing cholesterol concentration in the bile. Additionally, the sensitivity of the gallbladder to cholecystokinin owing to decreased cholesterol levels may promote gallbladder motility, which could prevent the occurrence of cholesterol polyps. GBP prevalence was significantly different between the two groups. We observed that the prevalence of GBPs varied by age, and the age distribution of residents differed between the two groups. In the JR group, a higher proportion of residents were younger than 60 years, which corresponded with an increased prevalence of GBPs. Conversely, the SR group comprised a larger percentage of participants aged 60 years or older, corresponding to a lower GBP prevalence. The age difference between the groups may explain the intergroup differences in the prevalence of GBP.

The age difference between the two groups can be illustrated by differences in the employment rates, lifestyles, and birth rates between the cities. According to a 2019 statistical report, Jeju City had 48,635 businesses (74%) compared with 17,463 (26%) in Seogwipo City [6]. Additionally, of 188 elementary, middle, and high schools on Jeju Island, 118 (62.7%) were in Jeju City [33]. People in their 40s or 50s, particularly those employed by businesses or with school-aged children, tend to favor residing in Jeju City. Conversely, Seogwipo City is the preferred residence for retirees engaged in farming or fishing. The distribution of residents aged below or above 60 years is likely influenced by the accessibility of essential services when selecting a place to live.

We found that the prevalence of GBPs increased until people reached their 50s in the JR group and started to decrease when they reached their 60s, which matched the findings of previous studies. However, in the SR group, GBP prevalence increased until people reached their 40s and then started decreasing thereafter. The reduced fat intake due to aging alone does not explain these results. High-risk alcohol drinkers in the SR group may have exerted an influence in this regard. In the SR group, the proportion of high-risk alcohol drinkers was significantly higher among residents in their 30s and 50s, and among residents in their 40s, there was a nonsignificant trend toward a higher proportion of high-risk drinkers in the SR group (*p* = 0.076). These findings support our hypotheses.

The population of Jeju Island was reported to be approximately 695,382 in 2020 [7], with Jeju City accounting for 505,759 residents and Seogwipo City for 189,803. About 73% of the island’s inhabitants were classified as Jeju citizens. In this study, the numbers of JRs and SRs were 16,090 and 5644, respectively. The proportion of JRs in this study was around 74%, aligning closely with their representation in the total population. Consequently, the percentages of study participants relative to the populations of the two cities did not differ significantly, suggesting that there was likely no effect of population disparity on the results.

In this study, the proportion of high-risk alcohol drinkers was significantly higher in the SR group than in the JR group. Additionally, the higher levels of aspartate aminotransferase and gamma-glutamyltransferase in the SR group served as evidence of prolonged alcohol consumption. Blood alcohol stimulates cholecystokinin secretion in the pancreas. Cholecystokinin increases gallbladder motility and reduces cholesterol polyp formation. The GBP prevalence in the subset of SRs with a higher rate of long-term alcohol consumption might have been even lower than that observed in the broader SR group. Although alcohol consumption may play a role in reducing the prevalence of GBP, it is also harmful. Therefore, a prospective study is warranted to investigate whether (relatively low levels of) alcohol consumption can mitigate GBP risk without causing hepatotoxicity and to elucidate the association between alcohol consumption and GBP prevalence in the two regions.

The quantity of pure alcohol in a serving can vary based on the type and volume of the alcoholic beverage, as well as the size of the beverage container. According to the WHO, a standard drink is a glass containing 10 g of pure alcohol. However, this standard is not available in South Korea. Since 2018, a standard Korean drink has been defined as 7 g of pure alcohol [21]. The Korean National Health and Nutrition Examination Survey (KNHNES), published annually by South Korea’s Ministry of Health and Welfare, characterizes high-risk drinkers as those consuming, on average, ≥seven glasses for men and ≥five glasses for women at least twice a week [21]. The WHO, however, considers high-risk drinking as the consumption of ≥five glasses for men and ≥four glasses for women, also twice weekly or more. The difference in the number of glasses between the two definitions of high-risk drinkers is due to differences in standard drink sizes between KNHNES and WHO. Nevertheless, the overall quantity of pure alcohol consumed is analogous between the two definitions, indicating that differing criteria between the KNHNES and WHO definitions did not lead to discrepancies.

Our study had several limitations. First, this study was based on questionnaires; therefore, recall bias or self-reporting bias may have occurred despite the authors’ efforts to minimize it. Second, this was a retrospective study conducted on patients who visited the Health Screening and Promotion Center at Jeju National University Hospital in Jeju City, potentially limiting older SRs’ participation. However, as the only university hospital on the island, Jeju National University Hospital is accessible via public transportation, facilitating SRs’ attendance. Collaboration with organizations located in Seogwipo City may be necessary to confirm our research findings. Third, our study did not collect data on GBP-related factors like smoking history, alcoholic fatty liver disease, and weight changes. Fourth, sonograms cannot be used to interpret the composition of GBPs; therefore, we could not compare risk factors that depend on variations in GBP characteristics. Finally, our study was limited by its retrospective and cross-sectional design. In the future, a multicenter, prospective, observational study is necessary to provide more robust evidence.

Although this study had several limitations, this is the first report on the difference in GBP prevalence in these two regions separated by a volcano on an island formed through volcanic activity. A prospective study including multiple health checkup centers on Jeju Island is necessary to verify our findings. Whether residential location can be an independent risk factor affecting the prevalence of GBP remains to be confirmed.

In conclusion, this study showed a significant difference in GBP prevalence among residents of the two regions of Jeju Island. Age and alcohol consumption might contribute to this difference; however, further prospective cohort studies are needed to confirm our findings.

## Figures and Tables

**Figure 1 jcm-13-02290-f001:**
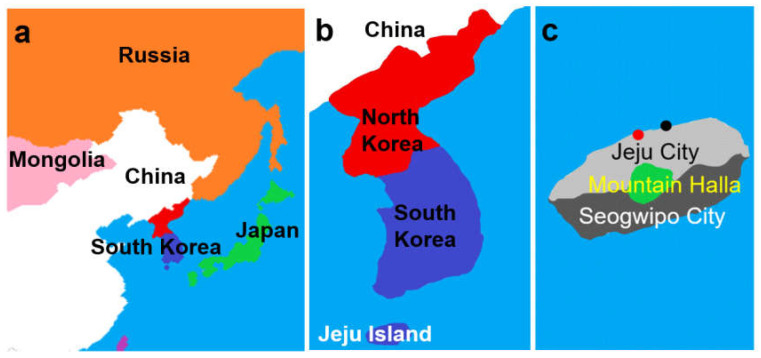
South Korea (**a**) is located in the Far East, surrounded by water (the Korean Peninsula). The middle figure (**b**) shows a map of South Korea. Jeju Island is 80 km south of the mainland of Korea. The right figure (**c**) depicts a map of Jeju Island, which is divided into two administrative regions including Jeju City and Seogwipo City, with Mount Halla at the center. The red dot indicates the location of Jeju International Airport, and the black dot marks the location of Jeju International Passenger Ship Terminal.

**Figure 2 jcm-13-02290-f002:**
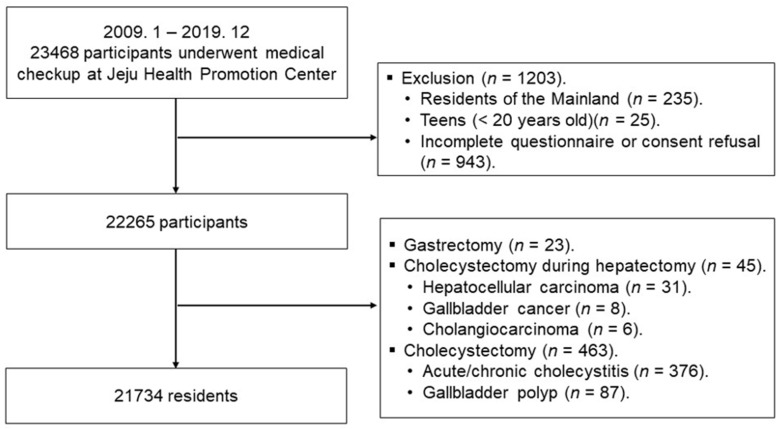
Flow diagram of included residents who underwent medical checkups.

**Figure 3 jcm-13-02290-f003:**
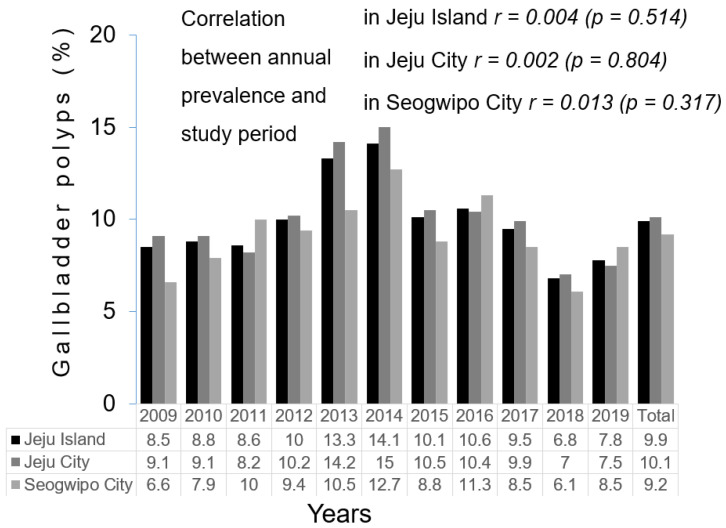
The annual prevalence of gallbladder polyps among residents who underwent medical checkups according to geographical location (Jeju Island, Jeju City, Seogwipo City) during the study period.

**Figure 4 jcm-13-02290-f004:**
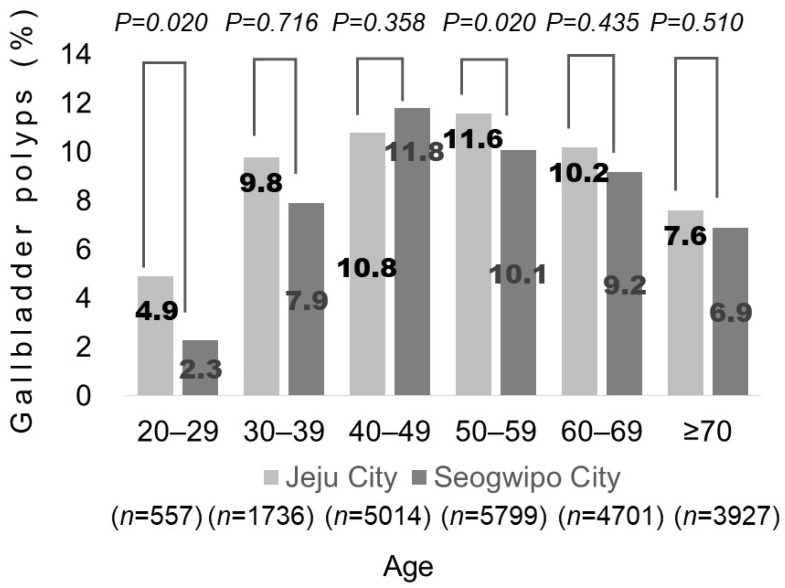
The prevalence of gallbladder polyps among residents who underwent medical checkups according to city (Jeju City and Seogwipo City) and age by decade (the 20–29, 30–39, 40–49, 50–59, 60–69, and ≥70 years age groups).

**Figure 5 jcm-13-02290-f005:**
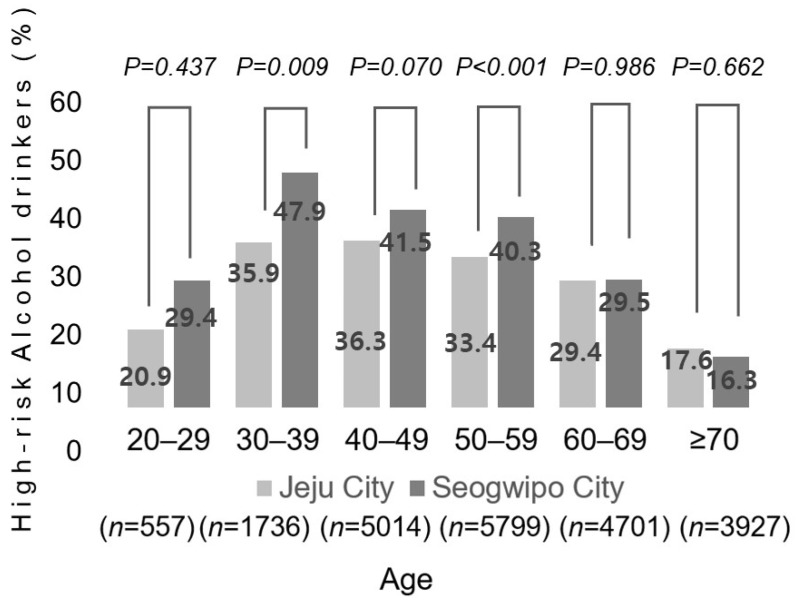
The rates of high-risk alcohol drinkers among residents who underwent medical checkups according to city (Jeju City and Seogwipo City) and age by decade (the 20–29, 30–39, 40–49, 50–59, 60–69, and ≥70 years age groups).

**Table 1 jcm-13-02290-t001:** Comparisons of clinical factors affecting gallbladder polyps among residents who underwent medical checkups in Jeju City vs. Seogwipo City, Jeju Island, Korea.

Variable	Jeju City (*n* = 16,090)	Seogwipo City (*n* = 5644)	*p*-Value
GBPs (%)	1577 (9.8)	502 (8.9)	0.043
Men (%)	8797 (54.6)	3075 (54.5)	0.804
Metabolic syndrome	2623 (16.3)	909 (16.1)	0.796
Central obesity ^a^	6243 (38.8)	2483 (44.0)	<0.001
High blood pressure ^b^	6484 (40.3)	2540 (45.0)	<0.001
Age (years)	54.5 ± 13.8	58.2 ± 14.1	<0.001
Body mass index (kg/m^2^)	24.6 ± 3.2	24.9 ± 3.2	<0.001
Fasting blood glucose (mg/dL), (70–110) ^c^	100.1 ± 31.1	101.6 ± 32.4	0.003
Total cholesterol (mg/dL), (<200) ^c^	195.3 ± 38.9	197.8 ± 39.2	<0.001
LDL cholesterol (mg/dL), (<100) ^c^	195.8 ± 38.7	198.1 ± 38.9	<0.001
HDL cholesterol (mg/dL), (>60) ^c^	53.6 ± 13.9	53.3 ± 13.9	0.191
Triglycerides (mg/dL), (<150) ^c^	121.1 ± 94.1	121.9 ± 106.8	0.637
AST (IU/L), (10–30) ^c^	28.8 ± 51.4	31.9 ± 102.1	0.003
ALT (IU/L), (10–40) ^c^	30.8 ± 67.1	32.6 ± 106.6	0.150
GGT (IU/L), (2–30) ^c^	50.9 ± 106.6	57.3 ± 130.3	<0.001
ALP (IU/L), (30–120) ^c^	215.5 ± 100.1	223.1 ± 115.2	<0.001
HBsAg	965 (6.0)	378 (6.7)	0.088
Physical activity	5165 (32.1)	1422 (25.2)	<0.001
High-risk alcohol drinker ^d^	5149 (32.0)	2003 (35.5)	0.005

Values are expressed as n (%) or mean ± standard deviation. ALP = alkaline phosphatase, ALT = alanine aminotransferase, AST = aspartate aminotransferase, BMI = body mass index, HBsAg = hepatitis B surface antigen, GGT = gamma-glutamyltransferase, HDL = high-density lipoprotein, LDL = low-density lipoprotein. ^a^ Central obesity was defined as a waist circumference ≥90 cm in men and ≥80 cm in women. ^b^ High blood pressure was defined as ≥130/85 mmHg. ^c^ Reference ranges are expressed in parentheses. ^d^ For men, a high-risk alcohol drinker was defined as a participant consuming ≥7 drinks of alcohol (≥5 drinks for women) and drinking two or more times per week. A Korean standard drink was defined as a portion containing 7 g of pure alcohol.

## Data Availability

The data are not publicly available due to there being no appropriate site for uploading them at present. The data presented in this study are available on request from the corresponding author.

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
