# Peer review of "Retrospective Study of the Prevalence and Associated Factors of Gallbladder Polyps among Residents of Two Korean Cities"

_jcm, 2024, doi:10.3390/jcm13082290_

Round 1

Reviewer 1 Report

Comments and Suggestions for Authors

Dear authors,

I have read the study titled “Prevalence and Associated Factors of Gallbladder Polyps Among Residents of Jeju City and Seogwipo City on Jeju Island, Korea, far from the Korean Peninsula: A Retrospective Cross-Sectional Study. The approach adopted in the study is very simplistic to study the participants of two regions based on the history of alcohol consumptions and taking of self-reported questionnaire-based collection of data. I have some comments which need to be addressed.

The title needs to be modified.

In the abstract section the authors have added extra unwanted details such” Jeju City has only an international airport and harbor. There are more businesses and schools in Jeju City than Seogwipo City. This kind of information are avoided in the summary section.

The key words should include the term alcohol consumption instead of region.

It is strange that the prevalence of gallbladder polyp is higher in Jeju City. However, the authors claim the it is due to alcohol consumption and alcohol consumption was higher in Seogwipo City.

The introduction last paragraph is similarly exaggerated with extra sentences and inclusion of figure which is not of much interest to the readers. The authors need to focus on the rationale behind the study rather than to write over exaggerated sentences in this part of the introduction.

Rs and SRs and not fully defined previously.

The authors have mentioned that the participants with high drinkers consumed seven or more glasses of alcohol and similar for another group. However, it is not clear what was the standard size of glass used in the consumption of alcohol. It is because it also varies with the amount of drink consumed. Similarly, it is not clear what sort of alcohol drink was utilized. Different brands with different sedation of alcohol are available in the market and each have varied effect on the body different from others. Like, suju a well-known Korean drink contain little amount of alcohol and one can drink many glasses of suju drink as compared to others where the alcohol quantity is much high. So, it is difficult to conclude that which category have shown the more pronounced effect.

In the results section the authors have mentioned figure 2 as percentages of the participants, However, it is no correct because figure 2 already have shown as flow chart in the method section.

It is very strange that the authors have taken the total participanats in included in the two cities not uniformly divided. The participants recruited and studied in Jeju are 16443 where it is 5, 806 only in the Seogwipo City. This kind of distribution is not justified b the conduct of the study.

The conclusion is not justified by the conduct of the study.

Comments on the Quality of English Language

Editing needed

Author Response

For research article

Response to Reviewer 1 Comments

1. Summary

I have read the study titled “Prevalence and Associated Factors of Gallbladder Polyps Among Residents of Jeju City and Seogwipo City on Jeju Island, Korea, far from the Korean Peninsula: A Retrospective Cross-Sectional Study. The approach adopted in the study is very simplistic to study the participants of two regions based on the history of alcohol consumptions and taking of self-reported questionnaire-based collection of data. I have some comments which need to be addressed

Response: We are thankful for your nice comments, which could make our study more scientific and logical. We have made every effort to address the reviewer’s questions point by point. We highlighted all manuscript revisions in red so that they can be easily reviewed by editors and reviewers.

2. Questions for General Evaluation

Reviewer’s Evaluation

Response and Revisions

Does the introduction provide sufficient background and include all relevant references?

Must be improved

We our best to provide sufficient background included all relevant reference.

Are all the cited references relevant to the research?

Can be improved

We add more references relevant to the research.

Is the research design appropriate?

Must be improved

The research design is improved than before.

Are the methods adequately described?

Must be improved

We make every effort to improve the method section.

Are the results clearly presented?

Must be improved

We modify the result section.

Are the conclusions supported by the results?

Must be improved

We modify the conclusion.

3. Point-by-point response to Comments and Suggestions for Authors

Comments 1: The title needs to be modified.

Response 1: Thank you for the insightful suggestion. We agree that the title could be improved. In the revised manuscript, we changed the title to: “Retrospective Study of the Prevalence and Associated Factors of Gallbladder Polyps among Residents of Two Korean Cities.” Look at line 2-3.

Comments 2: In the abstract section the authors have added extra unwanted details such” Jeju City has only an international airport and harbor. There are more businesses and schools in Jeju City than Seogwipo City. This kind of information are avoided in the summary section.

Response 2:  Thank you for the kind comment. We modified the sentence, which could have prejudiced readers, as you mentioned. Look at line 22.

Comments 3: The key words should include the term alcohol consumption instead of region.

Response 3: This study focused on the effect of alcohol consumption on the development or prevalence of GBP as a potential risk factor among residents. The keyword “region” should be replaced by “alcohol consumption”. Thank you for the nice comment. Look at line 38.

Comments 4: It is strange that the prevalence of gallbladder polyp is higher in Jeju City. However, the authors claim the it is due to alcohol consumption and alcohol consumption was higher in Seogwipo City.

Response 4: We are sorry to make the reviewer confused. In this study, the proportion of high-risk alcohol drinkers was significantly higher in the SR group than in the JR group. Additionally, the higher levels of aspartate aminotransferase and gamma-glutamyltransferase in the SR group served as evidence of pro-longed alcohol consumption. Blood alcohol stimulates cholecystokinin secretion in the pancreas. Cholecystokinin increases gallbladder motility and reduces cholesterol polyp formation. The GBP prevalence in the subset of SRs with a higher rate of long-term alcohol consumption might have been even lower than that observed in the broader SR group. We corrected the wrong sentence. Look at line 300 -306.

Comments 5: The introduction last paragraph is similarly exaggerated with extra sentences and inclusion of figure which is not of much interest to the readers. The authors need to focus on the rationale behind the study rather than to write over exaggerated sentences in this part of the introduction.

Response 5: As you mentioned, some sentences may confuse or misinform readers because they are long and contain slightly exaggerated expressions. This study was initially designed to prove the hypothesis that the prevalence of GBP may differ among residents of the two regions on Jeju Island, where the transportation of cargo and people is not favorable because of a high volcano. We needed to accurately describe the separation of the two regions. Therefore, characteristics describing the geography and the numbers of schools and occupations in Jeju Island were necessary for readers to fully understand the study’s hypothesis. Even though South Korea is well-known to foreigners, many do not know where Jeju Island is located or where it belongs. In addition, local residents of mainland South Korea also are unfamiliar with the regional characteristics of Jeju Island. Explanations with accompanying pictures are necessary. We added more information on Jeju Island and made the introduction section more logical. Thank you for your concern. See the introduction section in the revised manuscript.

Comments 6: Rs and SRs and not fully defined previously.

Response 6: The reviewer is concerned that SRs and JRs were poorly defined in the study and wanted detailed definitions of both. We added those to the methods section of the revised manuscript. Thank you for your nice comment. Look at line 106-110.

Comments 7: The authors have mentioned that the participants with high drinkers consumed seven or more glasses of alcohol and similar for another group. However, it is not clear what was the standard size of glass used in the consumption of alcohol. It is because it also varies with the amount of drink consumed. Similarly, it is not clear what sort of alcohol drink was utilized. Different brands with different sedation of alcohol are available in the market and each have varied effect on the body different from others. Like, suju a well-known Korean drink contain little amount of alcohol and one can drink many glasses of suju drink as compared to others where the alcohol quantity is much high. So, it is difficult to conclude that which category have shown the more pronounced effect.

Response 7: As the reviewer mentions, the amount of pure alcohol may vary depending on the type and volume of alcohol, as well as the size of the drinking glass. According to the World Health Organization (WHO), a standard drink is a glass containing 10 g of pure alcohol. However, that standard is not available in South Korea. Since 2018, a Korean standard drink has been a glass containing about 7 g of pure alcohol. Alcoholic beverages were categorized into 7 groups—soju (Korean distilled spirit), beer, makgeolli (rice wine), cheongju (clear refined rice wine), liquor, and other (including champagne and cocktails)—and the amount of pure alcohol was calculated depending on alcohol type. We could specify alcohol content as a percentage. For example, pure alcohol content was calculated as 4.5% for beer; 17.0%, soju; 6.0%, makgeolli; 12.0%, wine; 16.0%, cheongju; and 40.0%, liquor. Drinking glasses can differ based on the type of alcohol. Glass capacities were measured as 200 mL, beer; 50 mL, soju; and 200 mL, makgeolli; in South Korea. The specific gravity of alcohol was approximately 0.8 g/mL at 15°C. The content of pure alcohol could be calculated as: beer, 200 mL x 4.5 x 0.8 g/mL = 7.2 g/glass; soju, 50 mL x 17 x 0.8 g/mL = 6.8 g/glass; and makgeolli, 200 mL x 6 x 0.8 g/mL = 9.2 g/glass. According to a recent report on alcohol consumption trends in South Korea that used data from the 2018 Korea National Health and Nutrition Examination Survey (KNHNES), Koreans aged 19 or older consumed an average of 15.0 g of pure alcohol per day. Soju accounted for 64%, with 9.6 g/d; beer accounted for 24.5%, with 3.7 g/d; and makgeolli accounted for 5.7%, with 0.9 g/d. Together, beer and soju accounted for about 90% of alcohol consumption in South Korea. The Korean National Health and Nutrition Examination Survey (KNHNES), published annually by South Korea’s Ministry of Health and Welfare, characterizes high-risk drinkers as those consuming, on average, ≥ 7 glasses for men and ≥ 5 glasses for women at least twice a week. The WHO, however, considers high-risk drinking as the consumption of ≥ 5 glasses for men and ≥ 4 glasses for women, also twice weekly or more. The difference in the number of glasses between the two definitions of high-risk drinkers is due to differences in standard drink sizes between KNHNES and WHO. Nevertheless, the overall quantity of pure alcohol consumed is analogous between the two definitions, indicating that differing criteria between the KNHNES and WHO definitions did not lead to discrepancies. We are sorry to make the reviewer confused due to inaccurate sentences in the definition of high-risk drinker of method section. We modified and added those in the revised manuscript. Look at the Table 1 and line 116-117, 138-142 and 319-332. We are very thankful to the reviewer’s insightful comment.

Comments 8: In the results section the authors have mentioned figure 2 as percentages of the participants, However, it is no correct because figure 2 already have shown as flow chart in the method section.

Response 8: As the reviewer mentioned, figure 2 in the results section was incorrect. We corrected it with figure 3. Look at line 175. Thank you for the kind comment.

Comments 9: It is very strange that the authors have taken the total participanats in included in the two cities not uniformly divided. The participants recruited and studied in Jeju are 16443 where it is 5, 806 only in the Seogwipo City. This kind of distribution is not justified b the conduct of the study.

Response 9: The population of Jeju Island was reported to be approximately 695,382 in 2020, with Jeju City accounting for 505,759 residents and Seogwipo City for 189,803. About 73% of the island’s inhabitants were classified as Jeju citizens. In this study, the numbers of JRs and SRs were 16,090 and 5,644, respectively. The proportion of JRs in this study was around 74%, aligning closely with their representation in the total population. Consequently, the percentages of study participants relative to the populations of the two cities did not differ significantly, suggesting that there was likely no effect of a population disparity on the results. We described these points in the revised manuscript. Look at line 292-299. Thank you for nice comment.

Comments 10: The conclusion is not justified by the conduct of the study.

Response 10: This study aimed to show that the prevalence of GBPs was significantly different among residents who were divided geographically into two regions by a high volcano. We concluded that age and alcohol consumption could be contributing factors affecting GBP prevalence. However, as the reviewer mentioned, we drew our conclusion from a univariate analysis of clinical variables between the two groups instead of a multivariate analysis. We admit to the need to additionally confirm our findings via a further prospective cohort study. The authors agreed with the reviewer's opinion and modified the conclusion section to: “Age and alcohol consumption might contribute to this difference; however, further prospective cohort studies are needed to confirm our findings.” Thank you for your insightful comment.

4. Response to Comments on the Quality of English Language

Point 1:

Response 1: Response: This study needs to be proofread by a native English speaker and will be edited by the proofreaders provided on the official website. Thank you for the nice comment.

5. Additional clarifications

Reviewer 2 Report

Comments and Suggestions for Authors

The present manuscript is a large cohort retrospective study of the prevalence and factors associated with a rather controversial subject which is gall bladder polyps. 

The work had beautifully and smartly chosen 2 geographically similar but demographically different cities. As explained by authors; The phenotype of residents in those 2 cities is different hence the results can allow defining/correlating the prevalence of GBPs with different factors related to lifestyle and demography of the population in both cities.

Few comments should be highlighted:

Title is too long. another suggestion: "Retrospective study of the prevalence and associated factors of gall bladder polyps among residents of two Korean cities"

In the intro; the second paragraph entails that surgery is for GBPs that are >1cm or imposing medical burden. Yes, I do agree but GBP that is found to be increasing in size should also be added to the indications of surgery. Also what is meant by ethical issues arise when one refuses the operation?

Also in the introduction (and through the whole manuscript): the prevalence of GBPs is known to increase with age (not the opposite as mentioned in line 55). GBPs seen in children are beyond the scope of this manuscript and are due to other anatomical factors. This fact is important for interpreting the results. The sentence stating this was referenced to ref 7 (did not comment on the age) and ref 8 (did not find it). So, kindly revise this and revise the references. GBPs are known as stated by many large cohort studies to increase with age, if there are other studies saying the opposite they should be clearly put and the issue should be highlighted as a controversial issue.

In the results section 3.1: Two findings in this section were not explained or commented on in the discussion; 1)the large difference in the number of those coming for check ups between both cities. 2)the higher prevalence seen in 2014 and the lower prevalence seen in 2018.

The Table is not numbered, the values of different parameters (AST, ALT, etc...) should be associated with their reference normal values.

In the discussion: First, why all the correlation with diabetes and hyperlipidemia...etc. Why not simply say that the prevalence of GBPs correlated with the prevalence of MS? Second, kindly explain what is meant by the last sentence in p 6. "The lack of difference in prevalence was likely associated with a lack of changes in achievable abdominal ultrasonographic resolution throughout the period." Third, the sentence in line 268 should be modified to: lifestyle, occupations and birth rates. Fourth, there should be a comment on the 2 points I mentioned before (in the comments on the results in sec.3.1).

In the conclusion; The statistically significant difference in the prevalence of MS between both groups (being higher in SJ residents 16.6% vs 16.2% in SR residents) can explain the higher prevalence of GBPs in SJ residents. This should also be mentioned in the conclusion. 

Comments on the Quality of English Language

Minor adjustments:

Some words are not chosen rightly for the meaning. For example; In line 246: However is better than "of note". Also, the sentence in lines 268 and 269: explained is better than "illustrated", "bussinesses" should be replaced by occupation...etc.

Some sentences are too long so that they lost the correct meaning.

Author Response

For research article

Response to Reviewer 2 Comments

1. Summary

The present manuscript is a large cohort retrospective study of the prevalence and factors associated with a rather controversial subject which is gall bladder polyps. 

The work had beautifully and smartly chosen 2 geographically similar but demographically different cities. As explained by authors; The phenotype of residents in those 2 cities is different hence the results can allow defining/correlating the prevalence of GBPs with different factors related to lifestyle and demography of the population in both cities.

Response: We are thankful for the nice comments, which could make our study more scientific and logical. We have made every effort to address the reviewer’s questions point by point. We highlighted all manuscript revisions in red so that they can be easily reviewed by editors and reviewers.

2. Questions for General Evaluation

Reviewer’s Evaluation

Response and Revisions

Does the introduction provide sufficient background and include all relevant references?

Yes

Thanks

Are all the cited references relevant to the research?

-

We add more references relevant to the research.

Is the research design appropriate?

Yes

Thanks

Are the methods adequately described?

Yes

Thanks

Are the results clearly presented?

Yes

Thanks

Are the conclusions supported by the results?

Can be improved

We modify the conclusion.

3. Point-by-point response to Comments and Suggestions for Authors

Comments 1: Title is too long. another suggestion: "Retrospective study of the prevalence and associated factors of gall bladder polyps among residents of two Korean cities"

Response 1: Thank you for your insightful suggestion. The tile could be better. As the reviewer suggested, we changed the title in the revised manuscript to: “Retrospective Study of the Prevalence and Associated Factors of Gallbladder Polyps among Residents of Two Korean Cities.”

Comments 2: In the intro; the second paragraph entails that surgery is for GBPs that are >1cm or imposing medical burden. Yes, I do agree but GBP that is found to be increasing in size should also be added to the indications of surgery. Also what is meant by ethical issues arise when one refuses the operation?

Response 2: We wanted to express the surgeon's ethical regret over missing the appropriate opportunity for treatment. However, the sentence, “Ethical issues arise when one who has a GBP > 1 cm refuses surgery,” could be misunderstood, as the reviewer mentioned. We modified the sentence: “Additionally, medical concerns persist when a patient with a GBP > 1 cm refuses cholecystectomy, given the risk of cancer development before the next ultrasound exam.” Look at line 53.

Comments 3: Also in the introduction (and through the whole manuscript): the prevalence of GBPs is known to increase with age (not the opposite as mentioned in line 55). GBPs seen in children are beyond the scope of this manuscript and are due to other anatomical factors. This fact is important for interpreting the results. The sentence stating this was referenced to ref 7 (did not comment on the age) and ref 8 (did not find it). So, kindly revise this and revise the references. GBPs are known as stated by many large cohort studies to increase with age, if there are other studies saying the opposite they should be clearly put and the issue should be highlighted as a controversial issue.

Response 3: As you kindly mentioned, numerous cohort studies suggest that the prevalence of GBP increases with age. However, this finding is not universal, as some studies report conflicting results, leaving the association between advancing age and GBP prevalence as a subject of ongoing debate. The references you mentioned were cited because we needed to explain the association between alcohol consumption and the development or prevalence of GBPs. We condignly modified the introduction section. Look at line 72-5.

Comments 4: In the results section 3.1: Two findings in this section were not explained or commented on in the discussion; 1)the large difference in the number of those coming for check ups between both cities. 2)the higher prevalence seen in 2014 and the lower prevalence seen in 2018.

Response 4: 1) This study was conducted at an institution in Jeju City. As the reviewer mentioned, that location might not be easy for SRs to access. A discrepancy in number of residents between the two groups might be assumed. However, the percentages of the two cities’ populations as released by the Jeju Special Self-Governing Provincial Office and the residents’ numbers in this study were similar. Therefore, there is no difference in accessibility to Jeju National University Hospital between the two cities. We addressed the reviewer’s concerns in the revised manuscript. Look at line 292-9. 2) The highest and lowest prevalences of GBPs were observed in 2014 and 2018, respectively. We inferred that the variation was not attributable to changes in the actual prevalence of GBPs among residents or to the resolution of the ultrasound equipment but rather to the radiologists conducting the examinations. Ultrasound examinations were carried out by radiologists on an annual contract with the Health Screening and Promotion Center. The detection rate of GBPs could vary depending on the extent to which radiologists are encouraged to identify polypoid lesions or to distinguish GBPs from gallbladder stones, regardless of the ultrasound equipment’s resolution capabilities. Cross-validation is essential to minimize errors associated with interobserver variability. Look at line 237-245. Thank you insightful comment.

Comments 5: The Table is not numbered, the values of different parameters (AST, ALT, etc...) should be associated with their reference normal values.

Response 5: As the reviewer pointed out, the table was not numbered. We numbered it and added reference normal values ​​of different parameters to make the table easier to understand. Look at the Table 1.

Comments 6: In the discussion: First, why all the correlation with diabetes and hyperlipidemia...etc. Why not simply say that the prevalence of GBPs correlated with the prevalence of MS? Second, kindly explain what is meant by the last sentence in p 6. "The lack of difference in prevalence was likely associated with a lack of changes in achievable abdominal ultrasonographic resolution throughout the period." Third, the sentence in line 268 should be modified to: lifestyle, occupations and birth rates. Fourth, there should be a comment on the 2 points I mentioned before (in the comments on the results in sec.3.1).

Response 6: First, some parameters, such as central obesity, fasting blood glucose, total cholesterol, and LDL cholesterol, had significant associations with MS criteria. However, based on the definition of MS, the correlation between the proportion of residents with MS and GBP prevalence did not reach statistical significance in this study. Therefore, we could not state that GBP prevalence correlated with MS.

Second, the sentence the reviewer pointed out was translated from Korean to English, which can create ambiguity for the reader. The sentence was replaced with another readable sentence: “Even though residents’ medical checkups were performed using the latest ultrasound equipment available during the study period, a statistical increase in GBP prevalence was not observed. Therefore, GBP prevalence did not increase with ultrasound equipment resolution during the study period.” Look at line 232-6).

Third, we totally agree with the reviewer’s comment. We accepted the suggestion and modified the sentence. Thank you for your good comments.

Fourth, this study was conducted at an institution in Jeju City. As the reviewer mentioned, that location might not be easy for SRs to access. A discrepancy in number of residents between the two groups might be assumed. However, the percentages of the two cities’ populations as released by the Jeju Special Self-Governing Provincial Office and the residents’ numbers in this study were similar. Therefore, there is no difference in accessibility to Jeju National University Hospital between the two cities. We addressed the reviewer’s concerns in the revised manuscript. Look at line 292-9.

Comments 7: The statistically significant difference in the prevalence of MS between both groups (being higher in SJ residents 16.6% vs 16.2% in SR residents) can explain the higher prevalence of GBPs in SJ residents. This should also be mentioned in the conclusion. 

Response 7: We are so sorry to offer the difficult-to-read table. As a result, the reviewer misunderstood. Even though the proportions of residents with MS were different between the two cities, that difference did not reach statistical significance. We improved the readability of Table 1. Thank you for your comment.

4. Response to Comments on the Quality of English Language

Point 1: Minor adjustments: Some words are not chosen rightly for the meaning. For example; In line 246: However is better than "of note". Also, the sentence in lines 268 and 269: explained is better than "illustrated", "bussinesses" should be replaced by occupation...etc.

Response 1: The authors corrected a couple points that the reviewer mentioned. Thank you for the kind comment.

Point 2: Some sentences are too long so that they lost the correct meaning.

Response 2: This study needs to be proofread by a native English speaker and will be edited by the proofreaders provided on the official website. Thank you for the nice comment.

5. Additional clarifications

Round 2

Reviewer 1 Report

Comments and Suggestions for Authors

The authors have revise d the paper in light of my concern. I have nothing to add further.

Comments on the Quality of English Language

Okay